# Identification of 13 Novel Loci in a Genome-Wide Association Study on Taiwanese with Hepatocellular Carcinoma

**DOI:** 10.3390/ijms242216417

**Published:** 2023-11-16

**Authors:** Ting-Yuan Liu, Chi-Chou Liao, Ya-Sian Chang, Yu-Chia Chen, Hong-Da Chen, I-Lu Lai, Cheng-Yuan Peng, Chin-Chun Chung, Yu-Pao Chou, Fuu-Jen Tsai, Long-Bin Jeng, Jan-Gowth Chang

**Affiliations:** 1Center for Precision Medicine and Epigenome Research Center, China Medical University Hospital, Taichung 40447, Taiwan; orangeliu72@gmail.com (T.-Y.L.); 036019@tool.caaumed.org.tw (C.-C.L.); 025074@tool.caaumed.org.tw (Y.-S.C.); 092989@tool.caaumed.org.tw (Y.-C.C.); 035882@tool.caaumed.org.tw (H.-D.C.); vero0214@gmail.com (I.-L.L.); yunhun1013@gmail.com (C.-C.C.); yubau.bi04g@g2.nctu.edu.tw (Y.-P.C.); 2Million-Person Precision Medicine Initiative, Department of Medical Research, China Medical University Hospital, Taichung 40447, Taiwan; 3Department of Laboratory Medicine, China Medical University Hospital, Taichung 404, Taiwan; 4Department of Internal Medicine, Section of Hepatobiliary Tract, China Medical University Hospital, Taichung 40447, Taiwan; 010456@tool.caaumed.org.tw; 5Department of Medical Research, China Medical University Hospital, Taichung 40447, Taiwan; 6School of Chinese Medicine, China Medical University, Taichung 40402, Taiwan; 7Division of Pediatric Genetics, Children’s Hospital of China Medical University, Taichung 40447, Taiwan; 8Department of Medical Laboratory Science and Biotechnology, Asia University, Taichung 41354, Taiwan; 9Department of Surgery, Section of Hepatobiliary Tract, China Medical University Hospital, Taichung 40447, Taiwan; 010919@tool.caaumed.org.tw; 10Department of Bioinformatics and Medical Engineering, Asia University, Taichung 41354, Taiwan; 11Department of Medical Laboratory Science and Biotechnology, China Medical University, Taichung 40402, Taiwan

**Keywords:** biobank, chronic hepatitis, meta-analysis, polygenic risk score, PheWAS

## Abstract

Liver cancer is caused by complex interactions among genetic factors, viral infection, alcohol abuse, and metabolic diseases. We conducted a genome-wide association study and polygenic risk score (PRS) model in Taiwan, employing a nonspecific etiology approach, to identify genetic risk factors for hepatocellular carcinoma (HCC). Our analysis of 2836 HCC cases and 134,549 controls revealed 13 novel associated loci such as the *FAM66C* gene, noncoding genes, liver-fibrosis-related genes, metabolism-related genes, and HCC-related pathway genes. We incorporated the results from the UK Biobank and Japanese database into our study for meta-analysis to validate our findings. We also identified specific subtypes of the major histocompatibility complex that influence both viral infection and HCC progression. Using this data, we developed a PRS to predict HCC risk in the general population, patients with HCC, and HCC-affected families. The PRS demonstrated higher risk scores in families with multiple HCCs and other cancer cases. This study presents a novel approach to HCC risk analysis, identifies seven new genes associated with HCC development, and introduces a reproducible PRS model for risk assessment.

## 1. Introduction

Liver cancer is the fifth most common cancer and the second common cause of cancer deaths worldwide, and hepatocellular carcinoma (HCC) is the predominant type [1]. In addition to genetic factors, several other factors increase the risk of HCC, including chronic viral infection, alcohol abuse, diabetes mellitus (DM), obesity, metabolic diseases, hemochromatosis, and autoimmune hepatitis [1,2]. More than 70% of patients with HCC in Asia have a chronic hepatitis B virus (HBV) infection [2,3]. The lifetime risk of HCC is approximately 10–30% in individuals with chronic HBV infection [4]. The incidence of noninfectious HCC is increasing in developed countries due to the rising prevalence of obesity, DM, and metabolic diseases [5,6]. These risk factors lead to liver injury and progressive inflammation during which liver cells undergo cycles of necrosis and regeneration and, thus, develop somatic mutations and chromosomal instability [7,8]. Inherited liver disorders causing chronic inflammation, fibrosis, and cirrhosis can lead to the development of liver cancer. Because of the rarity and diversity of these disorders, the relative risk of HCC in these patients and the age at which tumors typically arise cannot be accurately estimated [9]. Approximately 3–5% of HBV-related HCCs demonstrate familial HCC aggregation that may result from genes with moderate or high penetrance in a population [10]. Multifactorial inheritance can increase the risk of HCC in those with a family history of liver cancer and lead to an earlier age at onset [11]. By contrast, patients with sporadic cancer typically have a later age at onset, likely due to interactions between hereditary and nonhereditary causes.

A genome-wide association study (GWAS) is a powerful method to explore genetic associations in HCC. Many GWASs have identified genetic factors associated with the development of HCC and numerous single-nucleotide variations (SNVs) in different genomic regions with potential importance in HCC susceptibility, including SNVs in the chromosomal regions 1p36.22, 2q32.3, 6p21.32, 6q15.21, 7q21.13, 8p12, 15q13.3, and 21q21.3 [12,13,14]. These studies have provided crucial insights into the genetic complexity of HCC. Previous studies have focused only on one specific etiology of HCC, especially viral-infection-related HCC, to determine genetic factors associated with HCC development. These studies did not explore complex interactions among genetic factors, environmental factors, and personal lifestyle changes in developing countries.

The implementation of various measures to prevent HBV- and hepatitis C virus (HCV)-related HCC since 1984 in Taiwan and the adoption of more aggressive approaches for managing HBV and HCV infection have resulted in a decline in the number of HBV carriers, thus reducing the prevalence of HBV-related HCC in the young population [15,16]. Under the national health program, Chinese herbs are prescribed for the treatment of various diseases, including chronic hepatitis, and these herbs may contain HCC-related aristolochic acid or liver toxins [17,18]. In addition, the booming economy has resulted in an increase in the prevalence of DM, metabolic disorders, and obesity [13,19]. Thus, due to these complex interactions, the genomic susceptibility of HCC in Taiwan may differ from other areas that were mainly caused by HBV, HCV, nonalcoholic fatty liver disease (NAFLD), or alcoholic liver disease.

We performed a GWAS by including large sample sizes of patients with HCC (*n* = 2836) and controls (*n* = 134,549) to identify novel loci for HCC and subsequently used them to conduct a polygenic risk analysis. Our results revealed many new HCC-associated loci and changes in the polygenic risk score (PRS) in the familial cancer group. The findings of this study enhance our understanding regarding the genetic susceptibility and development of HCC and provide new targets that can be considered for the prevention and treatment of HCC.

## 2. Results

### 2.1. Study Flowchart

Figure 1 presents the study design. After quality control, we obtained 508,004 variants for the 173,135 CMUH samples and 686,439 variants for the 88,347 TWB samples. We combined the genetic data of the two groups and then used Beagle 5.2 to impute more SNVs. We obtained 15,358,452 variants and 261,482 samples. After performing quality control based on the aforementioned parameters, we finally included 13,692,222 variants and 258,066 samples in the following analysis.

### 2.2. Demographic Characteristics of Patients

We used EMRs to select patients with HCC and controls. We used the *International Classification of Diseases*, *Ninth Edition*, *Clinical Modification* and *International Classification of Diseases*, *Tenth Edition*, *Clinical Modification* codes (155.0, 155.2, 197.7, C22.8, C22.9, C7B.02, V10.07, and Z85.05) associated with HCC to identify the case group. Individuals without a diagnosis of any cancer in our EMRs were included in the control group. We included 2836 patients in the case group and 134,549 individuals in the control group. The proportion of men (65.37%) was higher in the case group than in the control group. The average ages of the case and control groups were 65.5 (standard deviation [SD] = ±12.6) and 51.4 (SD = ±17.8) years, respectively. The average BMI values of the case and control groups were 27.2 (SD = ±5.4) and 25 (SD = ±4.7), respectively. Significant differences in sex, age, and BMI (Table 1) were noted between the groups. In terms of comorbidities, the prevalence of cirrhosis (41.78%) and diabetes (34.52%) was significantly higher in the HCC group than in the control group. In terms of infection status, 118 (4.16%) patients were infected with both HCV and HBV, 1239 (43.69%) patients were infected with HBV, 707 (24.93%) patients were infected with HCV, and 761 (26.83%) patients were not infected with either HCV or HBV (Table 1). The rate of HBV or HCV infection was significantly higher in the case group.

### 2.3. GWAS for Taiwanese Patients with HCC without a Specified Etiology

A whole-genome scan was performed for 2836 cases and 134,549 controls, and the heritability variance of the cases determined using genome-wide complex trait analysis and genome-based restricted maximum likelihood tools was 0.1139 [20]. The findings of the Manhattan plot indicated that 35 SNVs located on different chromosomes were significantly associated with HCC, and the most significant SNVs were located on chromosome 6 (Figure 2A). In addition, we performed a meta-analysis by using the data of the BBJ and UK Biobank participants and determined a significant difference between variants on chromosome 19 and 22, which demonstrated a weak association with HCC in our discovery GWAS study (Figure 2B) [21,22]. Furthermore, we subclassified these SNVs into four groups. Group 1 contained 13 novel SNPs (Table 2) with new loci that were identified only in this study, and the region plot of these new loci is presented in Appendix A. Group 2 consisted of SNVs that did not exhibit a significant association with HCC in the BBJ or UK Biobank but revealed a significant association in our meta-analysis (Appendix A). Group 3 consisted of SNVs that did not exhibit a significant association with HCC in our study but demonstrated a significant association with HCC in other meta-analyses (Appendix A). Group 4 consisted of SNVs that exhibited a significant association with HCC in our study and previous studies (Appendix A).

For group 1, seven of the thirteen novel-loci-related coding or noncoding genes, namely *F11/F11-AS1*, *PFKFB3*, *PRMT8*, *FAM66C*, *NAV2/NAV2-AS1*, *FRMD4A*, and *KIAA0232*, have been demonstrated to be correlated with the development of HCC or other cancers in many studies. Noncoding transcripts related to rs148610742 (*F11-AS1*) and rs150098717 (*FAM66C*) exhibited a function of decoy for miRNA regulation involving the development of HCC or other cancers [23,24,25]. The rs148610742-related gene *F11* exhibiting C8A/B complement binding was reported to be a prognostic predictor in HBV-infection-related HCC [26]. Genes related to rs77404202 (*NAV2*) and rs17155112 (*FRMD4A*) are involved in the Wnt/beta-catenin pathway for *NAV2* and the Hippo pathway for *FRMD4A*, respectively, which are HCC-development-related pathways [27,28]. The rs77404202 (*NAV2-AS1*) is a noncoding transcript that suppresses gene expression by binding complementary mRNA and inducing double-strand RNA degradation, such as in *NAV2* and *DBX1*. Genes related to rs117719091 (*PFKFB3*) and rs140233124 (*KIAA0232*) play crucial roles in cancer metabolism; *PFKFB3* induces glycolysis and activates hepatic stellate cells and then promotes liver fibrosis [29,30], and *KIAA0232* stimulates insulin secretion to regulate cancer metabolism [31,32]. The gene related to rs144225287 (*PRMT8*) encodes a protein arginine methyltransferase that can enhance cancer stem-cell function and cell proliferation [33,34]. The functions of the remaining six (rs187199523, rs144285059, rs6140450, rs74333160, rs80115676, and rs118180127) of the thirteen novel loci in HCC remain unknown. The rs187199523 locus has a binding site for the transcription factor ZKSCAN5 and is located on a related gene (*RP11-563D10.1*) intron, and *RP11-563D10.1* is a long noncoding RNA (lncRNA), which might be associated with cholangiocarcinoma [35]. The rs144285059 locus is located on the *LINC02511* intron, and *LINC02511* can modify the RNA of N6-methyladenosine-related lncRNA and, thus, predict the prognosis of ovarian cancer [36]. The rs74333160 locus has binding sites for various transcription factors, including ZNF354A and HNF1A/B, and the variant is located downstream of *AL355836.4*. The rs80115676 locus has binding sites for various transcription factors, including STAT2 and IRF1, and *RPL13P12* is the nearest gene. Rs6140450 is located on the Isl1 transcription factor binding site, and *RP1-209B9.2* is the nearest gene, which is a *HSPBAP1* pseudogene. For rs118180127, we discovered a transcription factor binding site of *ZNF282*, which is located 4 bp downstream of the variant, and *CTD-3023L14.3* is the nearest gene (Table 2). We used expression-quantitative trait loci (eQTL: GTEx_Analysis_v8 of liver tissue) to determine the effect of these loci on nearby genes, and the results revealed no eQTL for the six novel loci. Additionally, we conducted validation of the newly discovered 13 single nucleotide polymorphisms (SNPs) using an independent cohort (Case: 977; Control: 142,515) (Appendix A). Among these SNPs, only one (rs144285059) exhibited a statistically significant difference (Appendix A, *p* = 0.01009). In order to gain a better understanding of the disease associations of the 13 novel SNPs in our institution, we employed the pheWAS approach to explore the diseases associated with these SNPs in our patient population (Appendix A). Surprisingly, the pheWAS analysis revealed that 8 SNPs showed associations with various types of cancer, with a direct correlation between rs187199523 and hepatocellular carcinoma (HCC). For a detailed overview of the findings, please refer to Appendix A, where we present the diseases that achieved statistical significance after Bonferroni correction.

To evaluate the role of the 13 novel loci in the development of HCC, we analyzed differential gene expression in 34 noncancerous tissues and 71 HCC tissues by using edgeR. The results indicated that *KIAA0232* (*p* = 1.85 × 10^−8^) and *LINC02511* (*p* = 4.4 × 10^−4^) were significantly overexpressed and *F11* (*p* = 4.14 × 10^−3^), *FRMD4A* (*p* = 2 × 10^−2^), and *PFKFB3* (*p* = 1.06 × 10^−9^) were significantly under-expressed in the HCC tissues (Figure 3A–E). Moreover, we determined the clinical significance of these loci and observed that *F11-AS1* expression was correlated with poor survival in the patients with HCC (*p* = 0.034; Figure 3F; Appendix A.1,2).

We also used TCGA [37] and HCCDB (a database of hepatocellular carcinoma expression atlas) [38] to confirm our results. We noted that the differentially expressed genes exhibited similar expression trends in other databases (Appendix A). The expression of the rs148610742-related *F11/F11-AS1* gene was downregulated in TCGA and HCCDB, and the mutation rate of the *F11* gene was 5.2% (19/360) in TCGA. The downregulation of *F11-AS1* and *F11* was correlated with the survival of the patients with HCC in TCGA. The expression of the rs187199523-related *LINC02511* gene was noted in HCC tissues but not in noncancerous tissues in TCGA and normal liver tissues in the GTEx cohort. In addition, we observed that the rs187199523-related gene *RP11-563D10.1* was downregulated in the HCC tissues of the TCGA cohort. The rs77404202-related *NAV2* gene was upregulated in the HCC tissues of the TCGA cohort. The mutation rate of the *NAV2* gene was 8.6% (31/360) in TCGA–Liver Hepatocellular Carcinoma (LIHC) data, and the *NAV2* gene was reported to be associated with hepatitis but not HCC [39]. No mutations or expressions were observed in *LINC02511*, *F11-AS1*, *NAV2-AS1*, *DEFB109F*, and *RPL13P12* genes in the LIHC data in the TCGA (Appendix A).

For group 2, the loci-related genes *HLA-DPA2, HLA-DQA2*, *HLA-DQB1*, *HLA-DQB2*, *HLA-DQB3*, and *COL11A2P1*, were determined to be associated with HCC in the Taiwanese patients enrolled in this study but not in the UK Biobank and BBJ participants. We used the summary statistics of these studies and our data to perform a meta-analysis and found a significant association between these SNVs and HCC, indicating that these SNVs are unique for Taiwanese patients with HCC (Appendix A).

For group 3, the loci-related genes *IFNL3*, *IFNL4*, *HLA-DPA1*, and *HLA-DPB1* have been demonstrated to be associated with HCC in the UK Biobank and BBJ participants; however, no such association was observed in this study. We used the summary statistics of these studies and our data to perform a meta-analysis. The results revealed a decreasing association power between these SNVs and HCC, indicating that these SNVs are weakly associated with HCC in the Taiwanese population (Appendix A). We observed that the *p* values of the 71 SNVs were located at a significant border range (5 × 10^−8^ < *p* < 1 × 10^−5^) and found 25 SNVs that did not exhibit a significant association with HCC in this study, although these loci have been identified to play a crucial role in the development of HCC in other studies.

For group 4, our study and previous studies have reported an association with HCC, and the results revealed that most loci of this group were HLA-related SNVs. These HLA-related SNVs belonged to *HLA-DQB2* and *HLA-DPB1* (Appendix A). After combining our data with those of other studies to perform a meta-analysis, we identified that several SNVs for *HLA-DQ* and *COL11A2P1* were also associated with HCC in Taiwanese participants (Appendix A), and some subtypes of *IFNL3* and *IFNL4* exhibited a stronger correlation with HCC in Japanese participants (Appendix A).

### 2.4. Detailed Analysis of HLA Loci

Because HLA plays a crucial role in the development of HCC, we used imputation methods to subtype MHC class I and II to explore their association with HCC in the Taiwanese population. The allelic genotype of HLA genes was predicted using the HIBAG R package [40]. For MHC class I, the results revealed A*24:02 (*p* = 1.12 × 10^−7^, OR = 0.89, 95% CI = 0.85–0.93) and A*30:01 (*p* = 1.56 × 10^−7^, OR = 1.5, 95% CI = 1.28–1.76) for HBV infection (Appendix A.2), B*54:01 (*p* = 1.27 × 10^−4^, OR = 0.72, 95% CI = 0.61–0.85) for HCC (Appendix A.4), B*40:01 (*p* = 6.09 × 10^−7^, OR = 0.9, 95% CI = 0.86–0.94) and B*58:01 (*p* = 2.98 × 10^−17^, OR = 1.27, 95% CI = 1.2–1.34) for HBV infection (Appendix A.5), B*58:01 (*p* = 1.45 × 10^−3^, OR = 0.89, 95% CI = 0.82–0.96) for HCV infection (Appendix A.6), C*03:02 (*p* = 1.52 × 10^−16^, OR = 1.24, 95% CI = 1.18–1.31) and C*07:02 (*p* = 2.53 × 10^−6^, OR = 0.91, 95% CI = 0.88–0.95) for HBV infection (Appendix A.8), and C*0302 (*p* = 9.32 × 10^−5^, OR = 0.87, 95% CI = 0.82−0.93) and C*07:04 (*p* = 9.32 × 10^−3^, OR = 1.72, 95% CI = 1.12–2.78) for HCV infection (Appendix A.9). For MHC class II, *DPA1*, *DPB1*, *DQA1*, *DQB1*, and *DRB1* were associated with the development of HBV infection, HCV infection, or HCC, including DPA1*01:03 (*p* = 9.95 × 10^−5^, OR = 1.13, 95% CI = 1.06–1.2) and DPA1*02:02 (*p* = 8.18 × 10^−5^, OR = 0.9, 95% CI = 0.85–0.95) for HCC (Appendix A.10), DPA1*01:03 (*p* = 1.30 × 10^−86^, OR = 1.4, 95% CI = 1.35–1.45) and DPA1*02:02 (*p* = 2.24 × 10^−88^, OR = 0.73, 95% CI = 0.71–0.76) for HBV infection (Appendix A.11), and DPA1*01:03 (*p* = 1.29 × 10^−2^, OR = 0.94, 95% CI = 0.9–0.99) and DPA1*02:02 (*p* = 4.52 × 10^−2^, OR = 1.04, 95% CI = 1–1.09) for HCV infection (Appendix A.11). The top-10 risk or protection subtypes of MHC class I and II are listed in Table 3.

### 2.5. PRS Analysis of HCC in the Taiwanese Population

We divided GWAS data into three groups before the analysis: base, target, and validation. These three groups were considered to be independent samples. We used data from the base group to calculate summary statistics and then built a model using data from the target group. Finally, we used data from the validation group to verify the accuracy of the model (Figure 4 and Appendix A.1,2). The PRS distribution and statistical test results of the target group are presented in Figure 4A, and the PRS of the patients with HCC was significantly higher than that of the controls (Figure 4A, left, and Figure 4A, right). The odds ratio of PRS stratification with percentile is depicted in Figure 4B, and the results revealed an increase in the case-to-control ratio with progressively higher decile categories. Next, we confirmed the results using data from the validation group (Figure 4C). The PRS of the patients with HCC was higher than that of the controls (Figure 4C, left), and a significant difference in the PRS was noted between the patients with HCC and controls in the validation set (Figure 4C, right). We plotted the AUC to evaluate the performance of the PRS and determined that the PRS exhibited only a slight improvement in risk prediction. However, the addition of age, sex, BMI, albumin, HBV surface antigen, and HCV antibody as covariates considerably improved the performance of the prediction of HCC risk (Figure 4D). We also used the AUC to evaluate the performance of PRS with 20%, 15%, 10%, and 5% distribution, and the results indicated that a higher distribution of PRS exhibited better performance in terms of HCC-risk prediction (Appendix A, Appendix A.3). The forest plot of the odds ratios of covariates in the combined model demonstrated that the PRS had a higher odds ratio than other factors (Figure 4E). We predicted the risk of HCC based on the percentile of PRS and age stratification (Figure 4F). We observed that the risk of HCC progressively increased with age, although the patients had a similar PRS. In addition, we performed PheWAS analysis on the results of the polygenic risk score (PRS). We categorized the samples into two groups based on the PRS, with one group having scores greater than 90% and the other group having scores less than 10%. From the PheWAS results, we observed significant differences between patients with high PRS for hepatocellular carcinoma (HCC) and those with low PRS in terms of cancer of the liver and intrahepatic bile duct, viral hepatitis B, and viral hepatitis (Appendix A, Appendix A). This indicates that our PRS is able to identify individuals at higher risk for these diseases. The findings indicated that PRS only considers genomic information, but a disease onset is a complex condition that is affected by many factors including environmental changes, lifestyle, sex, and age.

### 2.6. PRS Analysis of the Family Members of Taiwanese Patients with HCC

To evaluate the effect of PRS on healthy individuals who have family members without cancer or with HCC or other cancer. The PRS distribution and statistical test results are shown in Figure 5 and Appendix A. The results revealed that the families with one member with other cancer (n = 3980) had the lowest average PRS, followed by the families with more than one member with other cancer (n = 1580), the families without a member with cancer (n = 11,665), the families with one member with HCC (n = 798), and the families with more than one member with HCC and other cancer (n = 560), and a significant difference was noted between the families without cancer and the families with more than one member with HCC and other cancer (*p* = 1.61 × 10^−3^; Figure 5A). We found no significant difference between the families with one member with HCC and the healthy families (*p* = 0.29; Appendix A, Figure 5A). However, when we rearranged these individuals into three groups as family members with HCC, with other cancer or without any cancer, a significant difference was noted between the families without a member with cancer and the families with at least one member with HCC (n = 1358; *p* = 5.9 × 10^−3^; Appendix A, Figure 5B).

## 3. Methods and Materials

### 3.1. Participants and Cohorts

We collected the details of one cohort including 88,347 participants from the Taiwan Biobank (TWB) [41] and another cohort including 175,997 participants from China Medical University Hospital (CMUH). The demographic data of the TWB cohort were collected from the TWB website (https://healthy.twbiobank.org.tw/ (accessed on 6 July 2021.)). The participants from CMUH were enrolled from three cohorts. Cohort 1 included patients enrolled in the Precision Medicine Project of CMUH that was initiated in 2018 and remained operational when this study was conducted. This project was performed to explore genetic factors associated with the development of common diseases in Taiwanese individuals and to develop a more precise system for predicting and preventing the occurrence of common diseases. This project mainly focuses on patients from CMUH. Cohort 2 consisted of patients whose data were collected from electronic medical records (EMRs) between 1992 and 2021, including their family history and laboratory data (e.g., DNA microarray results) for examining the side effects of drugs. The Department of Laboratory Medicine of CMUH (accredited by American College of Pathologists) uses the Taiwan Precision Medicine Initiative (TPMI) array to detect SNVs related to the side effects of drugs, including some crucial human leukocyte antigen (HLA) types, and this array contained 709,593 SNVs. Moreover, the TPMI array can be used to perform a GWAS of common diseases. Cohort 3 included the whole-genome sequencing and microarray data of patients enrolled in The Cancer Genome Atlas (TCGA) Sequencing project of CMUH (CMUH110-REC3-221). In this project, the genomes of more than 1000 patients with different types of cancers were sequenced using samples from the tissue bank of CMUH.

The Institutional Review Board (IRB) of the TWB (CMUH108-REC1-091) approved the inclusion of the TWB cohort. The IRB of CMUH (CMUH110-REC3-005) approved the inclusion of the cohort from the TPMI of CMUH. The IRB of CMUH (CMUH110-REC3-157) approved the collection of data from the EMRs of CMUH. We mixed the participants from the two cohorts in this study.

### 3.2. SNV Genotyping

Human genomic DNA was extracted from peripheral blood leukocytes by using a QIAamp DNA Micro Kit (Qiagen, Heidelberg, Germany) in accordance with the manufacturer’s protocol. The DNA concentration was quantified using the NanoDrop1000 spectrophotometer (Nanodrop Technologies, Wilmington, DE, USA) and a Qubit fluorometer (Invitrogen, Carlsbad, CA, USA). During the discovery phase, we genotyped 175,997 samples by using the TPMv1-customized SNV array (Thermo Fisher Scientific, Inc., Santa Clara, CA, USA), which was designed by the Academia Sinica and TPMI teams. The array contained approximately 714,431 SNVs. All the samples in our study had a call rate of >97%. To ensure the quality of SNVs, we excluded individuals and variants with missing rates (--geno 0.1 for variants and --mind 0.1 for individuals) and filtered out variants with a Hardy–Weinberg equilibrium *p* value of <10^−6^ (--hwe 1E-6) and a minor allele frequency (MAF) of <10^−4^ (--maf 0.0001), as determined using PLINK1.9 [42]. In addition, we removed the individuals by using principal component analysis (PCA); --pca) and heterozygotes (--het) as outliers. Finally, 508,004 variants and 173,135 individuals passed the filters and quality-control processes for autosomal chromosomes and were, thus, included in the subsequent analysis.

### 3.3. Phasing and Imputation Workflow

Before performing the imputation, we first constructed a haplotype reference panel and preprocessed SNV array data. From the whole-genome sequence (WGS) reference panels for the TPMI and TWB, we filtered out variants with a minor allele count (MAC) of <3, missing genotypes, multiple alleles (other than SNP/INDEL), and a Hardy–Weinberg equilibrium *p* value of <10^−7^ and then phased these reference panels by using SHAPEIT2 [43]. Using the pre-phasing WGSs of 1363 participants from the TWB as reference panels, we applied SHAPEIT4 to phase TPMI and TWB arrays. Finally, we performed imputation using Beagle5.2 [44], which is more effective and accurate than other imputation tools. The imputed data were filtered using an R-square alternate allele dosage of <0.3 and a genotype posterior probability of <0.9 as the criteria [45]. The SNPs present on the TPMv1 and TWB2 chips are identical; the discrepancy in naming arises from the chips being produced by different entities.

### 3.4. MHC Class I and Class II Allele Imputation and Subtyping

We developed an imputation method to predict HLA genotypes based on multiple SNVs present in the proximity of HLA regions and used this method to fine-map associated signals in complex regions. In this study, HLA imputation and model training were performed using HIBAG R package software [46]. HLAs were imputed using attribute BAGging, and SNV information was extracted from an extended MHC region ranging from 28,510,120 to 33,480,577 bp loci of chromosome 6 based on hg38 positions (6p21.3-22.1). Four-digit HLA imputation and typing were performed on *HLA-A*, *HLA-B*, *HLA-C*, *HLA-DRB1*, *HLA-DQA1*, *HLA-DQB1*, *HLA-DPA1,* and *HLA-DPB1* by using the Taiwanese population as reference. For post-imputation quality control, a call threshold of >0.9 was applied to remove poorly imputed HLA alleles [47].

### 3.5. GWAS

To determine associated variants, we used PLINK 1.9 to obtain the summary statistics of Taiwanese patients with HCC. We collected the data of patients who were diagnosed as having HCC in the EMRs of CMUH or the TWB and had data until the 4th follow-up questionnaire. In addition, we collected the data of controls who had no history of cancer and were aged >18 years. Finally, we included 2836 patients with HCC and 134,549 controls in the study. To determine the familial history of second-degree relatives, we obtained the data for only one person from a familial group for the targeted group but for both cases belonging to different phenotype groups. We determined the membership of the same family by using PLINK 2.0 Kindship. An additive genetic model is usually employed in case-control-based GWASs. Logistic regression was performed to analyze associations among traits after adjustment for multiple covariates (sex and age), and the most significant variant was selected to prevent a high level of collinearity in linkage disequilibrium (LD) that causes overestimation. The variant with a *p* value of <5 × 10^−8^ was considered to indicate a significant association between a case and a control. We plotted the Manhattan plot and quantile–quantile plot using an R package (“qqman”) and presented the region plot of the variants of interest by using LocusZoom tools [48].

### 3.6. Phenome-Wide Association Study (PheWAS)

We proceeded to perform subsequent PheWAS analysis on the newly discovered 13 SNPs in our study. Additionally, we conducted an analysis using the calculated polygenic risk score (PRS) for hepatocellular carcinoma (HCC). A total of 97,735,180 ICD-9 or ICD-10 diagnosis codes were collapsed into 1791 phecodes. The association between the PRS and each phecode was tested using logistic regression models and the “PheWAS” R package in R [49]. The PheWAS results were combined in a meta-analysis of multiple populations, with significance determined using Bonferroni correction.

### 3.7. Meta-Analysis

To validate variants detected from the summary statistics, we performed a fixed-effect meta-analysis based on three cohorts, namely the UK Biobank participants of European ancestry (154 cases and 420,117 controls; Pan-UKB team, https://pan.ukbb.broadinstitute.org, accessed on 30 November 2021) [50], the Japan Biobank (BBJ) participants of Japanese ancestry (1866 cases and 195,745 controls) [51], and the CMUH-TWB participants of Han Chinese ancestry. We used METAL with the sample size “effective N” as the weight for each cohort [52]. We transformed all the variants of the BBJ, UK Biobank, and our study to the rsID of dbSNP v.153 to determine significant variants before performing the meta-analysis to prevent the database difference between hg37 and hg38.

### 3.8. Statistical Analysis

We compared differences between the groups by using Student’s t test for the results of the digital data analysis and the chi-square test for the categorized data of clinical phenotypes (Table 1). Differential gene expression was analyzed, and the adjusted *p* value was evaluated using edgeR of the R package. Survival analysis was performed using the log-rank test. For the PRS distribution, we used the Mann–Whitney U test to analyze the PRS with z-score normalization between the cases and controls.

For the familial cancer study, 25,554 individuals with familial relationships were selected on the basis of the data of the third follow-up questionnaire of the TWB. Moreover, the Mann–Whitney U test was performed to evaluate the PRS in different target groups. The *p* value was adjusted using the false-discovery-rate Benjamini–Hochberg Procedure to prevent type I error, and an adjusted *p* value of <0.05 was considered statistically significant [53].

### 3.9. PRS Analysis

To calculate the PRS, we divided the CMUH cohort into three datasets: base, target, and validation. We used the base dataset to explore the association of the studied variables with HCC by using PLINK1.9 and then constructed a list of PRSs by using the target dataset and PRSice2 tools after filtering variants with a MAF of >0.01. We used the 1000 genome-phase v.3 of the East Asian population as a reference [54]. The PRS was calculated based on z-score normalization.

We used PRS, clinical data (including the albumin level, HBV surface antigen, and HCV antibody), or both to construct logistic regression models, and the last two models were adjusted by age, sex, and body mass index (BMI). Because extreme imbalance between cases and controls results in inflated performance, we used the oversampling method R “ROSE” package to eliminate this problem and validated the models through 10-fold cross-validation. Moreover, we used the validate dataset to confirm the PRS models.

### 3.10. RNA Sequencing Analysis of HCC Tumor Tissues

RNA was extracted from the tumor and adjacent noncancerous tissues of the patients with HCC by using TRIzol Reagent (Thermo Fisher Scientific, Inc., Santa Clara, CA, USA) or the NucleoSpin RNA kit (Macherey-Nagel, Takara Bio Inc., Kusatsu, Japan) in accordance with the manufacturer’s instructions. The quality of RNA (RNA integrity number, RIN > 6) was determined using the Agilent Bioanalyzer 4200 (Agilent Technologies, Santa Clara, CA, USA). One microgram of RNA and the TruSeq Stranded mRNA Library Prep kit (Illumina, San Diego, CA, USA) were used for library preparation in accordance with the manufacturer’s instructions. Briefly, total RNA was purified using magnetic beads to remove ribosomal RNA and fragmented through enzyme treatment. Subsequently, double-strand cDNA synthesis, end repair, adaptor ligation, and an enrichment polymerase chain reaction were performed. The samples were subjected to 2 × 150-bp paired-end sequencing using the Illumina NovaSeq 6000 platform (Illumina, San Diego, CA, USA). We used the DRAGEN bioinformatics workflow to analyze RNA sequences (Illumina DRAGENTM Bio-IT platform v3.7) and used gencode v35 as the gene model for RNA-read annotation.

## 4. Discussion

Most GWASs on HCC have focused on its unique etiology (such as HBV or HCV related HCC), including previous studies conducted in Taiwan [55,56], and these studies have explored only single-etiology-associated genetic loci. In this study, we adopted the mixed etiology approach to perform a GWAS for HCC, and this approach revealed not only complex interactions involved in the development of HCC in the Taiwanese population but also the association of loci with HCC identified in previous studies. In this study, we discovered many new associated loci and those already associated with other single etiologies. For example, rs2281293 (*PNPLA3*), a well-known SNV associated with alcoholic liver disease and NAFLD, was not found to be related to HBV- and HCV-related HCC in previous Taiwanese studies [55,57], but this association was observed in our study. The expression of this SNV was determined to be enhanced in our meta-analysis. Thus, our approach for performing a GWAS of HCC can be more suitable for exploring the associations between regions with complex interactions and HCC development.

In this study, we found 13 new loci, including sponge-like noncoding genes, liver-fibrosis-related genes, stem-cell-related genes, metabolism-related genes, and HCC-related pathway genes. The noncoding transcripts rs148610742 (*F11-AS1*) and rs150098717 (*FAM66C*) had a sponge-like function to prevent miRNA from targeting cancer-related genes involved in the development of HCC or other cancers [40,51,52]. The rs148610742 (*F11*) that binds with the C8A/B complement had been suggested as a prognosis marker in HBV-related HCC [41]. Rs77404202 (*NAV-AS1*) and rs144285059 (*LINC02511*) are lncRNAs that regulate target mRNA, and they were considered HCC-related lncRNAs in LncRNADisease v2. [58] Rs77404202 (*NAV2*) and rs17155112 (*FRMD4A*) have been shown in the Wnt/beta-catenin and Hippo pathway in HCC, respectively, and these pathways play crucial roles in HCC development [42,43]. The rs117719091-related gene *PFKFB3* can induce glycolysis and activate hepatic stellate cells to promote liver fibrosis, and the knockdown of *PFKFB3* inhibited HCC growth by damaging DNA repair function, leading to G2/M phase arrest and apoptosis [44,45]. The rs140233124-related gene *KIAA0232* affects the platelet count and insulin secretion and may play a role in the cancerous metabolism of HCC [46,47]. The rs144225287-related gene *PRMT8* controls embryonic stem cell pluripotency through the PI3K/AKT signaling pathway and may induce HCC progression through cell-cycle control [48,59]. The rs74333160 (*AL355836.4*), rs80115676 (*RPL13P12*), rs6140450 (*RP1-209B9.2*), rs187199523 (*RP11-563D10.1*), and rs118180127 (*CTD-3023L14.3*) are novel RNA or pseudogenes. These five loci-related genes may involve the development of HCC directly or indirectly through unknown mechanisms, which need further study. Compared with previous GWASs for HCC, this GWAS revealed several new findings, such as sponge-like noncoding genes and hepatic stellate cells, which induce liver fibrosis and play vital roles in HCC development. Our results may provide new approaches to prevent HCC development.

HLA plays crucial roles in the development of virus-related HCC [13,60,61,62,63,64]. We comprehensively analyzed the subgroups of MHC classes I and II and determined that class I plays a vital role in HCC development in the Taiwanese population, such as A*30:01 (*p* = 1.56 × 10^−7^, OR = 1.5, 95% CI = 1.28–1.76) for HBV infection, B*58:01 (*p* = 2.98 × 10^−17^, OR = 1.27, 95% CI = 1.2–1.34) for HBV infection, B*58:01 (*p* = 1.45 × 10^−3^, OR = 0.89, 95% CI = 0.82–0.96) for HCV infection, C*03:02 (*p* = 1.52 × 10^−16^, OR = 1.24, 95% CI = 1.18–1.31) for HBV infection, and C*03:02 (*p* = 9.32 × 10^−5^, OR = 0.87, 95% CI = 0.82–0.93) for HCV infection, and these results revealed that the MHC class I of the same subtype may have an opposite effect on different virus-related HCC (Appendix A).

In familial cancer studies, most researchers have focused on the family studies of breast and prostate cancer [65,66], and studies have rarely explored the familial association for HCC risk using PRS. We used GWAS-related SNVs to develop the HCC-related PRS and found the PRS can be used to identify individuals with a higher HCC risk in the general population, and the PRS can predict family members with a high risk of HCC. We also found that the PRS of the family with one member with HCC did not significantly differ from that of the healthy group (*p* = 0.29), indicating that not only the genotype but also other factors, such as environment and lifestyle, may play similar roles in the development of HCC. Interestingly, we found the family member with one or more other cancer had the lower PRS than that of the healthy group. The PRS model building based on the HCC cohort may not be suitable for predicting non-HCC cancer patients, and low PRS was only a marker for the risk of HCC but not for the risk of other cancer.

In this study, certain limitations warrant consideration. While we excluded individuals in the control group who had previously developed cancer, we were unable to effectively exclude patients in the control group who may have developed cancer outside of our institution or those who belong to high-risk populations for future HCC development. The precise categorization of high-risk factors for HCC, such as alcohol consumption, severity of fatty liver, and the extent of liver cirrhosis, was not achieved [67]. One significant contributing factor to this limitation is that finer categorization would result in reduced sample sizes, potentially affecting the statistical significance of genetic associations. In the future, our research will focus on individuals who have not been exposed to known risk factors but still develop HCC.

## 5. Conclusions

In this study, we encompassed a cohort comprising 2836 HCC cases alongside 134,549 matched controls. Our research has elucidated thirteen hitherto unidentified loci, with a minimum of seven implicated genes demonstrating associations with HCC or other neoplastic conditions. An extensive examination of the MHC subtypes revealed that certain subtypes are pivotal in the context of various viral etiologies and the pathogenesis of HCC. Utilizing PRS, we assessed the susceptibility of individuals with HCC, as well as those with familial ties to the disease. The insights gleaned from our investigation hold promise for the establishment of an innovative risk-stratification framework, aimed at forecasting HCC risk and susceptibility within families. The implications of our findings could potentially pave the way for novel preventative strategies against HCC.

## Figures and Tables

**Figure 1 ijms-24-16417-f001:**
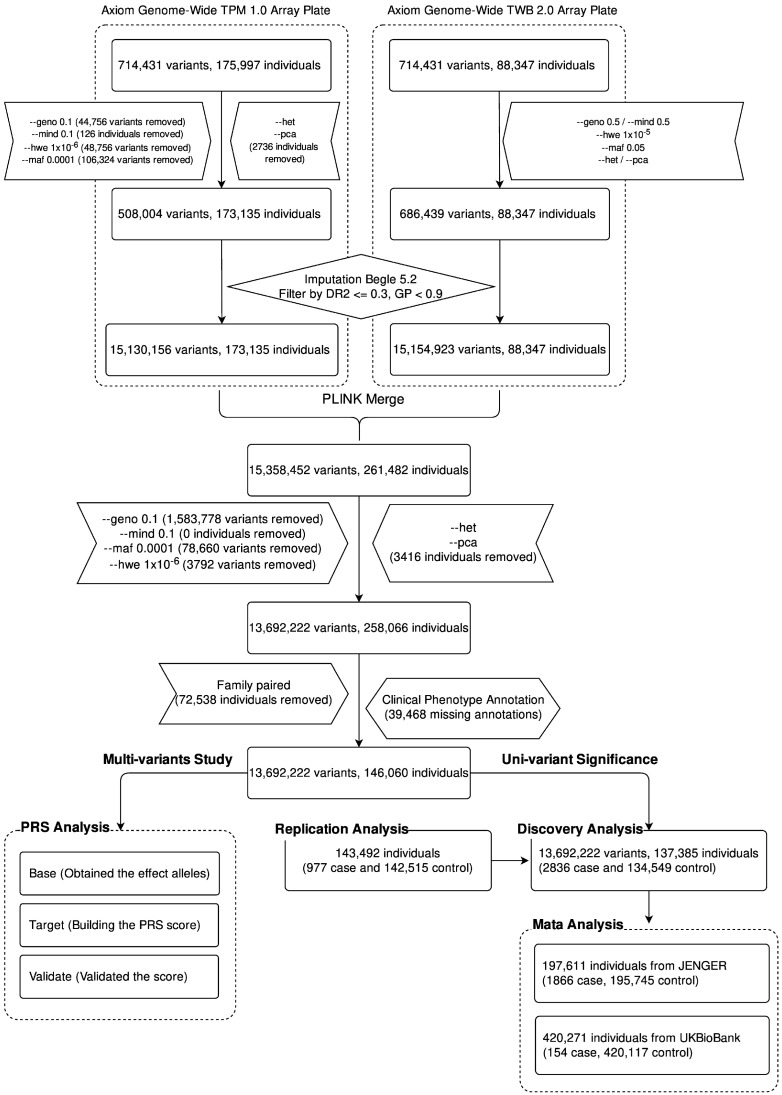
Flowchart of the construction of the GWAS and the calculation of PRS in Taiwanese patients with HCC. Commands prefixed with double hyphens signify instructions executed via the PLINK software suite, allowing replication of our analytic outcomes. The “geno” option assesses the missing rate of variants, while “mind” evaluates the missing rate among participants. The “hwe” command checks for Hardy–Weinberg equilibrium compliance. “maf” filters for minor allele frequency, and “pca” executes principal component analysis. “het” analyzes the heterozygote ratio. Following this quality control sequence, we employed Beagle 5.2 for imputation analysis. Alleles with a dosage r-squared (DR2) of less than or equal to 0.3 and a genotype posterior probability (GP) of less than 0.9 are excluded from subsequent analysis.

**Figure 2 ijms-24-16417-f002:**
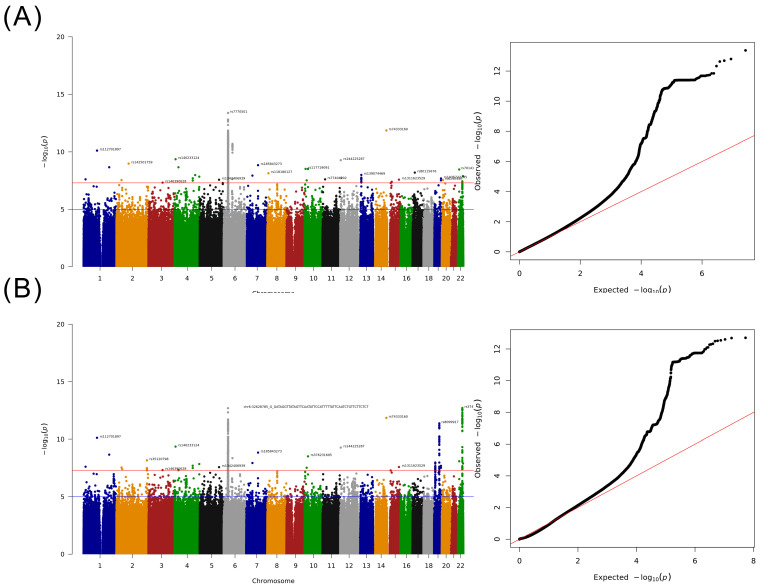
Results of the GWAS for Taiwanese patients with HCC. (**A**) Manhattan plot adjusted for sex and age and QQ-plot. (**B**) Manhattan plot of the meta-analysis study including BBJ and UK Biobank summary statistics and QQ-plot.

**Figure 3 ijms-24-16417-f003:**
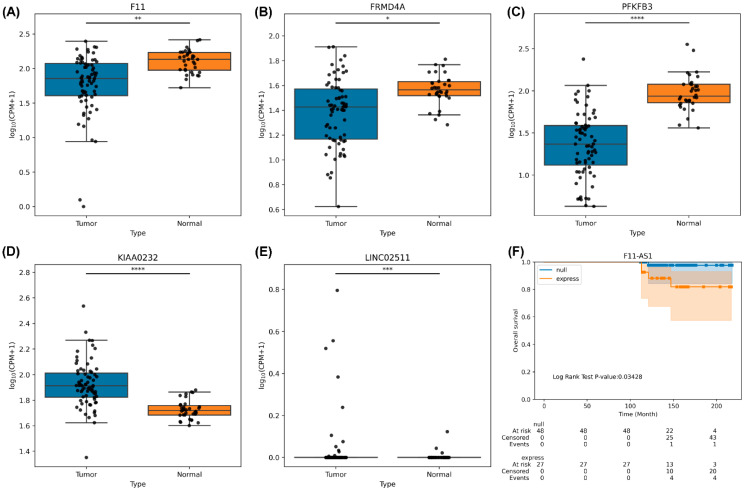
Results of novel SNP loci-related gene expression and survival between HCC and noncancerous tissues collected from Taiwanese patients with HCC and from TCGA: (**A**) F11 gene expression. (**B**) FMRD4A gene expression. (**C**) PFKFP3 gene expression. (**D**) KIAA0232 gene expression. (**E**) LINC02511 gene expression. (**F**) F11-AS1 survival analysis. *: 0.01 < *p* ≤ 0.05; **: 10^−3^ < *p* ≤ 0.01; ***: 10^−4^ < *p* ≤ 10^−3^; ****: *p* ≤ 10^−4^

**Figure 4 ijms-24-16417-f004:**
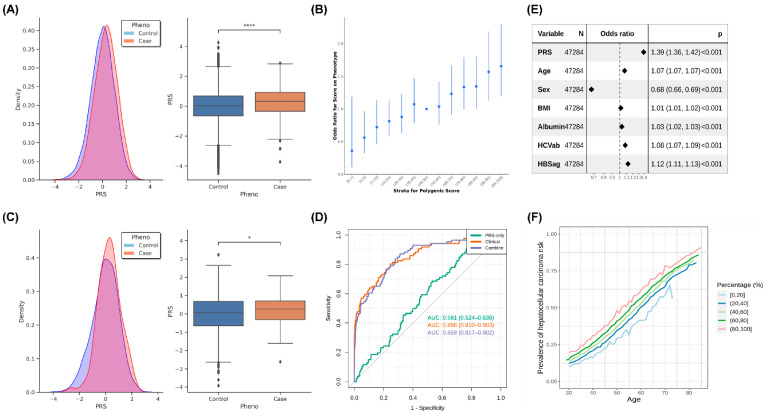
Results of PRS analysis: (**A**) The PRS distribution and statistical results of the target group. Left, the PRS distribution (the X axis: normalized PRS, Y axis: density); right, the statistical results for HCC cases and controls. (**B**) The odds ratio of PRS stratification with percentile. The 40–50 percentile was used as a reference to compare other groups to calculate the odds ratio, and 95% CI is shown as a line. (**C**) The PRS distribution and statistical results of the validate group. Left, the PRS distribution (the X axis: normalized PRS; Y axis: density); right, the statistical results for HCC cases and controls. (**D**) The ROC curve of the validate group. PRS only: only PRS was used in modeling; combined: using PRS, demography (age, sex, and BMI), and bioclinical data (Albumin, HBV surface antigen, and HCV antibody) in model building. (**E**) The forest plot of the odds ratios of covariants in the combined model with 95% CI. (**F**) Predicted HCC cancer risk by quintile stratification of HCC cancer PRS with increase in age. *: 0.01 < *p* ≤ 0.05; ****: *p* ≤ 10^−4^. “N” is the number of samples, and “*p*” is the statistical value (Odds ratio, 95% CI, *p*-value).

**Figure 5 ijms-24-16417-f005:**
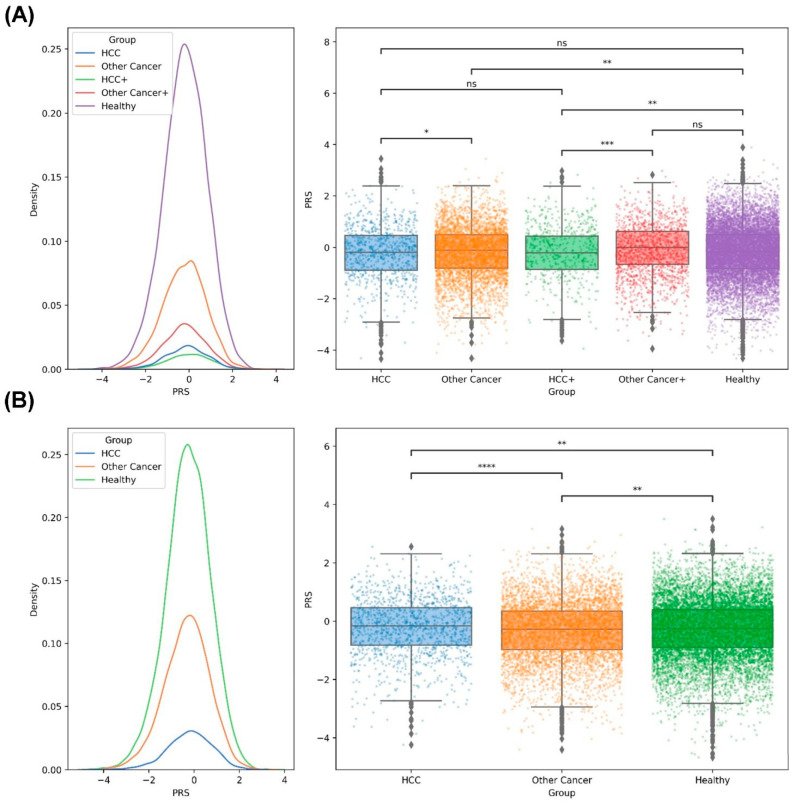
Results of PRS distribution and statistical difference between the members of the cancer family and noncancer family: (**A**) The distribution of PRS in different groups (left); the significant difference between groups is shown (right). HCC: one person in the family with HCC; Other Cancer: one person in the family with cancer (not HCC); HCC+: more than one person in the family with HCC and other cancers; Other Cancer+: more than one person in the family with cancer (not HCC); Healthy: no member in the family with cancer. (**B**) The distribution of PRS for HCC, other cancer (non-HCC), and healthy groups (left); the statistical difference between the different groups is shown (right). *: 0.01 < *p* ≤ 0.05; **: 10^−3^ < *p* ≤ 0.01; ***: 10^−4^ < *p* ≤ 10^−3^; ****: *p* ≤ 10^−4^; ns: not significant.

**Table 1 ijms-24-16417-t001:** Clinical characteristics of case and control in discovery GWAS cohort.

Variables			Case	Control	*p*-Value ^c^
Demography					
	Sex	male, n (%)	1854 (65.37%)	54,622 (40.60%)	3.9 × 10^−155^
		female, n (%)	970 (34.20%)	79234 (58.89%)	
		Unknown, n (%)	12 (0.42%)	693 (0.52%)	
	Age	year, mean (SD)	65.5 ± 12.6	51.4 ± 17.8 ^b^	0.0 × 10^+00^
	BMI	kg/m^2^, mean (SD)	27.2 ± 5.4	25.0 ± 4.7	4.6 × 10^−118^
		Unknown, n (%)	160 (5.64%)	19,104 (14.20%)	
Liver Disease					
	Cirrhosis ^a^	n (%)	1185 (41.78%)	93 (0.07%)	0.0 × 10^+00^
		Unknown, n (%)	1651 (58.22%)	134,456 (99.93%)	
	Virus Infection	HBVsAg(+), n (%)	1239 (43.69%)	8503 (6.31%)	0.0 × 10^+00^
		HCV(+), n (%)	707 (24.93%)	2118 (1.57%)	0.0 × 10^+00^
		Unknown, n (%)	246 (8.67%)	44,558 (33.12%)	
		HBVsAg(+) HCV(+), n (%) ^d^	118 (4.16%)	211 (0.16%)	
		HBVsAg(−) HCV(−), n (%)	761 (26.83%)	75,528 (56.13%)	
Metabolism					
	Diabetes	Type II diabetes, n (%)	979 (34.52%)	12461 (9.26%)	0.0 × 10^+00^
		HBVsAg(+), n (%)	139 (4.90%)	285 (0.21%)	
		HCV(+), n (%)	294 (10.37%)	205 (0.15%)	
		HBVsAg(+) HCV(+), n (%)	22 (0.78%)	5 (0.00%)	
		diabetes (others), n (%)	0 (0.00%)	246 (0.18%)	
		Non-diabetes, n (%)	1697 (59.84%)	102,747 (76.36%)	
		Unknown, n (%)	160 (5.64%)	19,095 (14.19%)	
		Total	2836	134,549	

The table show detail number of case and control group in demography, chronic liver and metabolism diseases. The gray color represents the subgroup of above condition (considering two conditions).Abbreviation: SD, Standard deviation; HBVsAg, Hepatitis B virus antigen; HCV, Hepatitis C virus. ^a^ Cirrhosis positive (cirrhosis had been diagnosised in CMU hospital electronic medical record system). ^b^ The Age < 18 were removed from our study. ^c^ Statistical significance of the difference between case and control group were calculated by chi-square test in the category or student t-test in digit, respectively. ^d^ The sample size considered HBV and HCV together, which sample included part of the HBV and HCV group.

**Table 2 ijms-24-16417-t002:** Summary statistics of 13 significant novel variants were found in discovery GWAS.

Marker	Variant			MAF			Discovery		Nearest Gene
	Chr	Position	RA/EA	PAF ^a^ (%)	Case (AF, %)	Control (AF, %)	OR (95% CI)	*p*-Value ^b^	
rs187199523	1	194027489	A/T	2.48	5672 (4.60)	283,510 (3.05)	1.49 (1.30–1.69)	2.17 × 10^−9^	RP11-563D10.1 (ENSG00000227240) *
rs140233124	4	6834347	A/-	6.05	5664 (7.20)	257,060 (4.91)	1.40 (1.26–1.55)	4.34×10^−10^	KIAA0232 *
rs144285059	4	136895711	-/A	4.29	5672 (4.76)	272,166 (3.19)	1.44 (1.27–1.64)	1.99 × 10^−8^	LINC02511
rs148610742	4	186288789	C/T	3.22	5668 (3.42)	276,154 (2.12)	1.54 (1.33–1.79)	1.43 × 10^−8^	F11 */F11-AS1 *
rs118180127	8	8513430	T/A	5.31	4894 (1.94)	260,472 (3.90)	0.55 (0.44–0.67)	7.30 × 10^−9^	CTD-3023L14.3 (ENSG00000253343)
rs117719091	10	6227313	C/T	4.05	4932 (0.85)	264,072 (2.66)	0.40 (0.29–0.54)	3.02 × 10^−9^	PFKFB3 *
rs17155112	10	14357172	G/A	2.77	5526 (2.01)	273,888 (1.22)	1.74 (1.43–2.12)	3.02 × 10^−8^	FRMD4A *
rs77404202	11	20117743	C/T	3.87	5144 (1.32)	268,176 (2.91)	0.50 (0.39–0.64)	2.46 × 10^−8^	NAV2 */NAV2-AS1
rs144225287	12	3568611	G/-	4.01	5668 (5.51)	269,502 (3.59)	1.46 (1.29–1.64)	5.27 × 10^−10^	PRMT8
rs150098717	12	8198462	C/T	3.43	5040 (0.83)	266,258 (2.25)	0.41 (0.30–0.56)	1.81 × 10^−8^	FAM66C/DEFB109F
rs74333160	14	101184577	T/G	5.21	5664 (6.94)	257,812 (4.51)	1.47 (1.32–1.64)	1.42 × 10^−12^	AL355836.4 (ENSG00000288245)
rs80115676	17	17375355	A/G	6.25	5014 (1.76)	264,032 (3.61)	0.53 (0.43–0.66)	6.34 × 10^−9^	RPL13P12 *
rs6140450	20	7873320	T/C	5.11	5016 (1.42)	264,980 (3.14)	0.51 (0.40–0.65)	2.79 × 10^−8^	RP1-209B9.2 (ENSG00000277315)

Abbreviation: Chr, chromosome; RA, reference allele; EA, effect allele; AF, allele frequency; PAF, publish allele frequency; OR, odds ratio; CI, confidence interval; MA, meta-analysis; MAF, minor allele frequency. ^a^ effect allele frequency in East Asian of gnomAD v3.1.2. ^b^
*p*-value was adjusted by the sex, age. * It is mean that the gene is expressed in normal liver tissue from GTEx Analysis Release V8.

**Table 3 ijms-24-16417-t003:** The top 10 risk and protect of HLA subtype association with different traits.

Trait	Risk			Protect		
	HLA-Type	OR (95% CI)	*p*-Value ^a^	HLA-Type	OR (95% CI)	*p*-Value ^a^
HCC	DQA1*04:01	1.71 (1.22–2.46)	3.30 × 10^−3^	B*54:01	0.72 (0.61–0.85)	4.19 × 10^−3^
	DQB1*04:02	1.69 (1.21–2.45)	4.33 × 10^−3^	DRB1*14:54	0.72 (0.62–0.84)	2.22 × 10^−4^
	DRB1*06:09	1.53 (1.21–1.96)	5.75 × 10^−4^	DQA1*06:01	0.74 (0.68–0.80)	1.16 × 10^−10^
	DRB1*13:02	1.52 (1.21–1.94)	8.28 × 10^−4^	DRB1*12:01	0.75 (0.65–0.87)	9.88 × 10^−4^
	DPB1*04:02	1.41 (1.05–1.92)	9.03 × 10^−2^	DRB1*12:02	0.78 (0.71–0.85)	7.90 × 10^−7^
	DQB1*06:02	1.38 (1.17–1.63)	4.08 × 10^−4^	B*38:02	0.81 (0.70–0.94)	8.79 × 10^−2^
	DQB1*03:02	1.35 (1.21–1.51)	4.43 × 10^−7^	DQB1*03:01	0.81 (0.76–0.86)	4.34 × 10^−10^
	DQA1*03:01	1.29 (1.14–1.47)	2.66 × 10^−4^	DQA1*01:04	0.85 (0.77–0.95)	8.45 × 10^−3^
	DRB1*15:01	1.24 (1.10–1.40)	9.61 × 10^−4^	DPB1*05:01	0.87 (0.82–0.92)	5.21 × 10^−5^
	DQA1*01:02	1.17 (1.08–1.26)	3.35 × 10^−4^			
HBV infection	DRB1*13:01	5.45 (3.14–10.35)	2.21 × 10^−14^	DPB1*05:01	0.69 (0.67–0.72)	1.56 × 10^−102^
	DQB1*06:03	4.60 (2.53–9.34)	9.95 × 10^−10^	DRB1*14:54	0.70 (0.64–0.76)	1.48 × 10^−16^
	DRB1*13:02	3.48 (2.93–4.17)	1.12 × 10^−63^	DQA1*06:01	0.73 (0.69–0.76)	5.18 × 10^−38^
	DQB1*06:09	3.45 (2.90–4.14)	1.30 × 10^−62^	DPA1*02:02	0.73 (0.71–0.76)	8.94 × 10^−88^
	B*44:03	2.33 (1.46–3.95)	6.49 × 10^−4^	DPB1*19:01	0.76 (0.67–0.86)	2.09 × 10^−05^
	DPB1*09:01	2.09 (1.72–2.56)	8.39 × 10^−16^	DQB1*03:01	0.76 (0.74–0.79)	2.22 × 10^−54^
	DPB1*17:01	1.90 (1.54–2.37)	1.45 × 10^−10^	DPB1*13:01	0.77 (0.72–0.82)	3.61 × 10^−15^
	DQB1*03:02	1.89 (1.77–2.03)	6.72 × 10^−85^	DRB1*12:02	0.78 (0.74–0.82)	5.66 × 10^−23^
	DRB1*01:01	1.84 (1.37–2.52)	3.11 × 10^−5^	DQA1*01:04	0.81 (0.76–0.85)	6.36 × 10^−14^
	DQA1*03:01	1.80 (1.67–1.95)	2.22 × 10^−57^	DQB1*03:03	0.82 (0.79–0.86)	7.67 × 10^−22^
HCV infection	C*07:04	1.72 (1.12–2.78)	5.90 × 10^−2^	DPB1*104:01	0.48 (0.27–0.91)	8.26 × 10^−2^
	C*08:01	1.14 (1.03–1.26)	7.51 × 10^−2^	DRB1*07:01	0.81 (0.70–0.95)	7.87 × 10^−2^
	DPB1*13:01	1.13 (1.02–1.25)	7.41 × 10^−2^	DRB1*13:02	0.82 (0.71–0.95)	5.97 × 10^−2^
	DQB1*03:01	1.10 (1.04–1.16)	2.12 × 10^−3^	DQB1*06:09	0.83 (0.72–0.97)	7.61 × 10^−2^
	DPB1*05:01	1.06 (1.01–1.12)	7.73 × 10^−2^	DRB1*03:01	0.86 (0.79–0.93)	4.01 × 10^−3^
	DPA1*02:02	1.04 (1.00–1.09)	9.04 × 10^−2^	DQB1*02:01	0.87 (0.81–0.94)	4.22 × 10^−3^
				C*03:02	0.87 (0.82–0.93)	1.77 × 10^−3^
				DQA1*05:01	0.88 (0.82–0.95)	2.32 × 10^−2^
				B*58:01	0.89 (0.82–0.96)	4.78 × 10^−2^
				DPB1*02:02	0.90 (0.82–0.99)	9.99 × 10^−2^

^a^ The *p*-value had been adjusted with fdr-bh, and the significance threshold was 0.1.

## Data Availability

The original contributions presented in the study are included in the article/Appendix A. Further inquiries can be directed to the corresponding authors. The data supporting the findings of the study are available to bonafide researchers upon approval of an application to the UK Biobank (https://www.ukbiobank.ac.uk/researchers/), Pan-UKB team (https://pan.ukbb.broadinstitute.org, accessed on 30 November 2021), and Japan Biobank (http://jenger.riken.jp/en/, accessed on 4 November 2021). Here is a link to the discovery association test study on LocusZoom: https://my.locuszoom.org/gwas/376115/?token=a04af13e1e9842568de52d860f84a25f. Here is a link showing the association test study via meta-analysis on LocusZoom: https://my.locuszoom.org/gwas/116439/?token=a70fa914666447488be9cc4ddbbf1b7f.

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
