# Peer review of "Identification of 13 Novel Loci in a Genome-Wide Association Study on Taiwanese with Hepatocellular Carcinoma"

_ijms, 2023, doi:10.3390/ijms242216417_

Round 1

Reviewer 1 Report

Comments and Suggestions for Authors

The authors provide a very powerful study on HCC with a very large cohort. They could clarify some of the methodological details. What is the difference between the two versions of the array. How they calculated the PRS score should be provided in details such as beta values with snp mafs? did they add these?  Do they have any correlation with survival, treatment outcome etc? Additionally, how does the SNPs relate to gene expression. did they see any of these SNPs in TCGA?

Author Response

Dear Reviewer,
Thank you for acknowledging the significance of our study on hepatocellular carcinoma (HCC) within the substantial Taiwanese cohort. We are thankful for the opportunity to elucidate the methodological aspects you have queried. Please find a more detailed response in the attached document.

Reviewer 2 Report

Comments and Suggestions for Authors

The article “Identification of 13 Novel Loci in a Genome-wide Association Study on Taiwanese with Hepatocellular Carcinoma” by Liu et al. is an interesting study. The authors performed a genome-wide association study in Taiwan using a non-specific etiology model to identify genetic risk factors for hepatocellular carcinoma (HCC). It presents a new approach to HCC risk analysis, identifies new genes associated with HCC progress, and brings into use for the first time a reproducible polygenic risk score (PRS) model.

This article has some shortcomings that need to be removed, corrected, and improved, please see below:

1.     The title is not correctly written in all-caps title style! Please, correct it!

2.     The abstract is too short and does not provide enough information about the study design, results, and conclusions. It could be improved. Please, revise and improve the text.

3.     Some keywords are already in the title and should not have been included as keywords once again. Please, adopt the MeSH system for choosing the keywords and revise them accordingly.

4.     Writing the references and citing them in the text is NOT uniform. Please, review all references to be cited correctly and uniformly.

Please, see for example:

Line 49: “…is the predominant type.[1]

Please, be careful and correct, the point must be placed after closing the parenthesis, for e.g., [1].

Line 52: hemochromatosis, and autoimmune hepatitis.[1, 2] Here are represented as superscript!

And the examples can go on throughout the entire article!!!

5.     The introduction should be revised, and more references should be added to support the study.

6.     Please, explain in Fig. 1 by a Legend all abbreviations used so that the reader can understand the diagram (the flowchart).

7.     All figures and tables should be better explained and discussed. All figures and diagrams should be redesigned in a blind palette of colors.

8.     The final discussions and conclusions should be improved.

9.     This article necessarily requires a graphical abstract for the reader to better understand the scientific content.

10.  References must be checked and rewritten in the style required by IJMS and MDPI platform.

References should be double-checked and improved with a digital object identifier (DOI), in MDPI style. For example, http://dx.doi.org/10.9993/ajae/aaq93, which will take the reader directly to the information page for the article.  

11.   Academic editing of English, grammar, and style is required.

Overall, I recommend a minor revision.

I believe that after this revision provided by the authors on the issues suggested to be corrected and improved, it will provide useful and credible information for all readers, especially researchers. It is up to the Academic Editor to decide on its publication.

Thank you very much!

03 November 2023

Comments on the Quality of English Language

Academic editing of English, grammar, and style is required.

Author Response

(The authors gave the same response as above.)
